# Effects of Prone Positioning on Head Control in Preterm Infants: Randomized and Controlled Clinical Trial Protocol

**DOI:** 10.3390/ijerph20032375

**Published:** 2023-01-29

**Authors:** Sabrinne Suelen Santos Sampaio, Nathalia Allana de Amorim Rodrigues, Julia Raffin Moura, Carolina Daniel de Lima-Alvarez, Silvana Alves Pereira

**Affiliations:** 1Post-Graduation Program of Physiotherapy, Universidade Federal do Rio Grande do Norte (UFRN), Natal 59078-970, Rio Grande do Norte, Brazil; 2Departament of Physiotherapy, Universidade Federal do Rio Grande do Norte (UFRN), Natal 59078-970, Rio Grande do Norte, Brazil; 3Post-Graduate Program in Sciences of Rehabilitation, Universidade de Brasília (UNB), Brasília 72220-275, Distrito Federal, Brazil

**Keywords:** newborn, premature birth, tummy time, prone position, infant development

## Abstract

Purpose: The primary aim will be to assess the effects of prone positioning (tummy time) on cervical extension (angular kinematics and time) in preterm infants. The secondary aim will be to assess the effects of tummy time on gross motor function. Methods: This randomized, controlled clinical trial will include 40 preterm infants weighing less than 2500 g, randomly allocated into control or experimental group (n = 20) and followed up from birth to six months of corrected age by the team of the neonatal follow-up clinic. Caregivers will be routinely guided on bonding, developmental milestones, and how to perform the tummy time for 30 min throughout the day (experimental group). An illustrative booklet will be provided as support material. The hypothesis will be tested using inferential analysis, considering an alpha of 5%. Discussion: We expect tummy time to strengthen cervical muscles needed to overcome gravity, master motor skills, and stimulate the integration between family activity and environmental experiences, considerable challenges to which preterm infants are exposed. Trial registration: Registered in the Brazilian Registry of Clinical Trials (identifier RBR-2nwkr47) on 17 February 2022.

## 1. Introduction

The American Academy of Pediatrics [1], using the Back to Sleep educational campaign [2], recommended avoiding prone positioning when sleeping to reduce the number of sudden deaths in newborns in the United States. This practice spread worldwide and reduced the number of unexpected mortalities. However, the prolonged use of the supine position delayed neuromotor development [3,4] and increased the cases of cranial asymmetry (e.g., positional plagiocephaly and brachycephaly) [5]. Therefore, caregivers began to be encouraged in the use of prone position when infants were awake to reduce morbidities associated with the supine position and increase play and family bonding [6,7]. This practice, also known as tummy time [8], was then disseminated worldwide through the educational campaign Back to Sleep, Tummy to Play [9].

Tummy time is also encouraged by the World Health Organization (WHO) [10] for infants under one year of age, possibly becoming the first opportunity for physical activity. Although Brazil does not have a specific recommendation on tummy time, the Brazilian Society of Pediatrics follows the recommendations of the American Academy of Pediatrics [11] and the WHO [10]. Tummy time can be performed starting at birth, strengthening the head, neck, and trunk muscles needed to overcome gravity and master motor skills (e.g., rolling over, sitting, crawling, and switching positions) [4,12]. Two systematic reviews found an association between tummy time and the prevention of positional skull deformity [3,13], which delays motor development. Studies also indicated that this delay might be due to limited head movement [14] or cranial alteration in brain formation [15].

Studies established a relationship between tummy time and motor development in infants with typical and atypical development [12,16]. These effects may also extend to preterm infants at risk of developmental delays due to immaturity of the neurological [17] and musculoskeletal systems [18].

Premature birth hinders essential and organizing experiences needed for good intrauterine development, possibly leading to changes in brain maturation and negatively impacting the interaction between caregiver and infant. Moreover, family engagement and empowerment are essential to improve the environment and development of a preterm infant [19].

Tummy time can notably integrate family activity and environmental experiences, considerable challenges to which preterm infants are exposed [20,21]. Mendres-Smith et al. [7] observed that tummy time conducted by parents decreased negative vocalizations and increased exercise tolerance in infants. Thus, this study hypothesizes that tummy time will provide adequate motor development in preterm infants, especially for motor skills in the first months of life. The primary aim will be to assess the angular kinematics of the cervical extension; the assessment of gross motor function will be the secondary aim.

## 2. Materials and Methods

### 2.1. Study Setting and Participants

This randomized, controlled, and double-blinded clinical trial will be reported according to the Consolidated Standards of Reporting Trials (CONSORT) [22] and the Template for Intervention Description and Replication checklist (TIDieR) [23].

Data will be collected at the Pediatrics and Childcare outpatient clinic of the Januario Cicco Maternity School of the Federal University of Rio Grande do Norte (Natal, Rio Grande do Norte, Brazil). The sample will comprise preterm infants born in the maternity school, followed up at the clinic, and recruited via direct contact with the caregivers by the researchers.

### 2.2. Sample Size

The sample size was calculated using the G Power software, considering the Ancova test. The randomized controlled pilot study of Flierman et al. [24] was considered to determine the effect size (Cohen’s d). The estimated minimum sample size was 40 preterm infants, considering the gross motor subtest of 9.1 ± 1.7 and 8.1 ± 2.0 (Cohen’s d of 0.59), two groups, four assessments, sampling error of 0.05, and a power of 0.80.

### 2.3. Inclusion Criteria

We will include preterm infants born between 30 and 36 weeks and 6 days (confirmed by ultrasound); with adequate gestational age; Apgar ≥ 7 in the 5th minute; birth weight < 2500 g; without cardiorespiratory, neurological, musculoskeletal, and auditory complications; without genetic syndromes, congenital infections, and visual changes; and with a poor or unusual motor repertoire in the assessment of generalized movements.

### 2.4. Exclusion Criteria

We will exclude preterm infants diagnosed with grade III and IV intraventricular hemorrhage and those whose caregivers withdraw. After the randomization, we will exclude preterm infants who present any worsening of health and need hospitalization, those who move from the state, who do not attend two evaluations, and those whose caregivers do not answer the phone for three consecutive days to reschedule the assessments.

### 2.5. Randomization and Blinding

Randomization will be stratified by gender and preterm infants will be separated into two groups using a computer-generated randomization sequence [25]. Three researchers, not involved in the study, will be responsible for allocation and assessments of the infants and for data analysis. The allocation will be sealed in opaque envelopes at the assignment and will keep the randomization confidential during the study. The assessments results will not be disclosed to the researcher responsible for the interventions and the data will be tabulated with a predetermined encoding by another blinded evaluator.

Another researcher not involved in the study and blinded to the allocation will conduct all assessments; results will not be disclosed to the researcher responsible for the interventions. The parents will be instructed, previously, not to tell the researcher what kind of orientation they are receiving. Figure 1 illustrates the protocol according to Standard Protocol Items: Recommendations for Interventional Trials (SPIRIT) [26].

### 2.6. Measures

#### Sample Characterization Measures

A structured form will be used, with questions about prenatal (gestational and maternal age, number of prenatal consultations, pregnancies, deliveries, abortions, and antenatal corticosteroid use), perinatal (birth weight, Apgar in the fifth minute, and delivery conditions), and postpartum (weight and gestational age at hospital discharge, days of hospitalization and complications, and use of mechanical ventilation and oxygen therapy).

### 2.7. Outcome Measures

#### 2.7.1. Primary Outcome

Assessments of angular kinematics and time for cervical extension will be conducted using the Kinovea^®^ software version 0.9.5 (Joan Charmant & Contributors, Bordeaux, France), which will perform a 2D kinematic analysis of movements recorded during cervical extension in the prone position [24]. This software has adequate reliability, validity, and reproducibility for analyzing dynamic movements [27,28,29]. Goniometry for cervical flexion (0 to 65°) and extension (0 to 90°) will determine the points and plot the angles of cervical extension [30].

Time for cervical extension will be considered when the preterm infant starts cervical extension at 0° and reaches a maximum elevation between 45° and 90°. 

#### 2.7.2. Secondary Outcome

Gross motor function will be assessed using the Bayley Child Development Scale—III, a gold standard instrument to assess child development. This scale was translated, validated, and adapted for Brazilian children [31]. Standard and composite scores will consider a mean of 10 ± 3 of standard deviation (SD) and 100 ± 15, respectively. Following the qualitative classification of the manual [31], the delay threshold will be 79 points (1.5 SD below the mean), easing the comparison with the expected percentages on the normality curve.

### 2.8. Assessment Procedures

#### 2.8.1. Sample Selection

The sample will be selected at the pediatrics and childcare outpatient clinic. The researcher responsible for the selection will invite all caregivers of preterm infants who meet the inclusion criteria.

#### 2.8.2. Follow-Up

After the first outpatient consultation, a researcher will allocate the preterm infants into control or experimental group and follow-up with them during routine consultations; this researcher will be blinded to the allocation. All preterm infants will also be followed up by telemonitoring. The researcher responsible for the intervention will contact all caregivers weekly to answer questions, schedule evaluations, and encourage participation.

#### 2.8.3. Study Groups

In the control group, caregivers will routinely receive guidance from the care team of the follow-up outpatient clinic (e.g., doctors, nurses, and physical therapists) about bonding and developmental milestones, following the Child’s Handbook [32].

In the experimental group, caregivers will receive the same guidance as the control group and an illustrative booklet on tummy time. The advice will be verbally reinforced to recommend tummy time for 30 min daily throughout the day to 3 months of corrected age [10].

Caregivers will be instructed to conduct and supervise activities while the preterm infants are alert and awake. The preterm infant must be placed on a fixed and comfortable surface to perform the tummy time, and the caregivers must pay attention to signs of tiredness (e.g., crying or resting the face on the surface), ending the practice before the infant gets tired. 

The booklet will be delivered on the first day and will comprise an illustrative guideline on the exercises and the proper type of surface, and a space to record the practice time. The booklet was based on guidelines from the American Occupational Therapy Association [33] (Appendix A).

#### 2.8.4. Assessment

A research team will be responsible for assessments and will be blinded to allocation. Four assessments will be performed: one pre-intervention (baseline) and three post-intervention. The baseline assessment will be at hospital discharge (beginning of the intervention), and the next will be performed at 2, 3, and 4 months of corrected age. Figure 2 shows the study schema.

#### 2.8.5. Data Analysis

Data analysis will be performed by a researcher blinded to allocation, using the Statistical Package for the Social Sciences software (IBM, Armonk, NY, USA). Only the primary investigator (not involved in data collection) may stop the study and have access to the final data.

The descriptive analysis will be presented as measures of central tendency and dispersion, and the inferential analysis will consider a statistical significance of 5%. The Shapiro–Wilk test will analyze data distribution. Moreover, the split-plot ANOVA will be used to compare differences between groups and assessments, and linear mixed models will analyze possible differences between groups. Regarding the interventions, the *t*-test for independent samples or the Mann–Whitney test will be used for intergroup comparisons.

Group description will be presented as mean and SD, and the effect size and 95% confidence interval will be reported. The intent-to-treat analysis will be performed for missing data, considering the last data of the preterm infant.

## 3. Discussion

This protocol proposes an intervention with the participation of the parents in the care of infants’ development. The participation of the parents provides better long-term results for the child and improves family well-being [34,35], as the opportunity for interacting with the environment, through the family context, decisively influences motor development [21]. Studies have shown that, even with motor alterations caused by prematurity, early interventions and stimuli provided by the family have positive effects on infants’ motor development [19,35]. The main components that contribute to the participation of the families are the psychosocial support for parents and parental education [35]. Psychosocial support contributes to the reduction of stress, anxiety, and depressive symptoms and increases parental sensitivity and responsiveness, as the empowerment of child care and parental education guides the ability of parents to provide better care for premature babies by feedback from professionals that support the assistance [35].

The strength of the study described here is the encouragement of family participation using a low-cost intervention that can be started right after hospital discharge and, thus, can promote the parents’ empowerment in the care of their children, optimizing the development of preterm infants. With the quantitative analysis of head control and motor development, we can effectively analyze the benefit of the intervention and, thus, obtain scientific support to encourage it, empowering parents in the care of preterm babies from the hospital environment to home.

## 4. Conclusions

Prematurity is one of the main public health problems in Brazil. Early motor interventions and the involvement of parents in the first months of life becomes urgent. Family engagement and empowerment are important strategies to improve the environment and the performance of preterm infants. The adoption of a simple and low-cost early intervention practice, such as tummy time exposure, conducted by caregivers at home, has a great potential for improving motor development in preterm infants.

## Figures and Tables

**Figure 1 ijerph-20-02375-f001:**
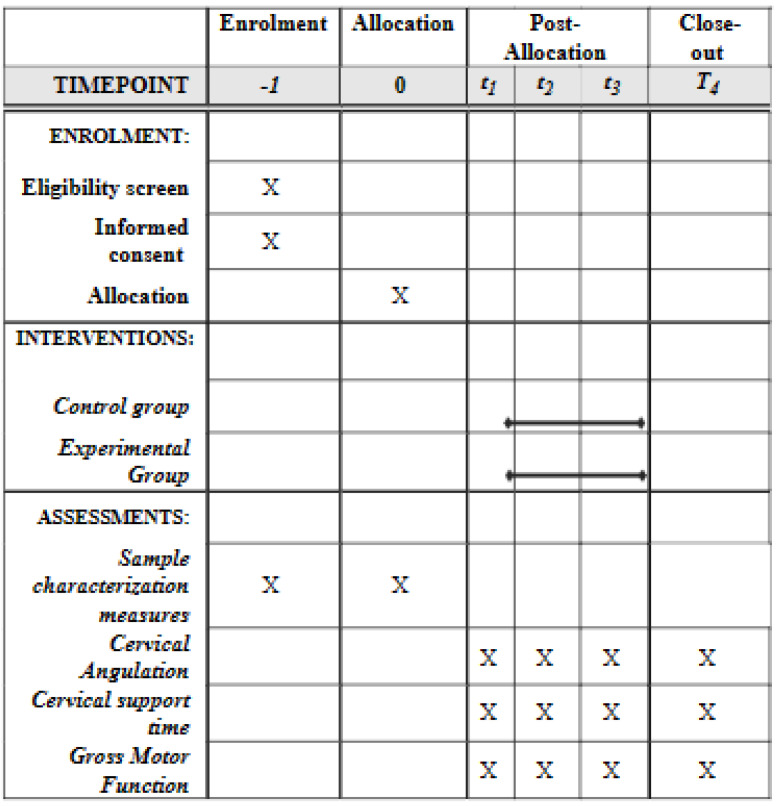
Study protocol.

**Figure 2 ijerph-20-02375-f002:**
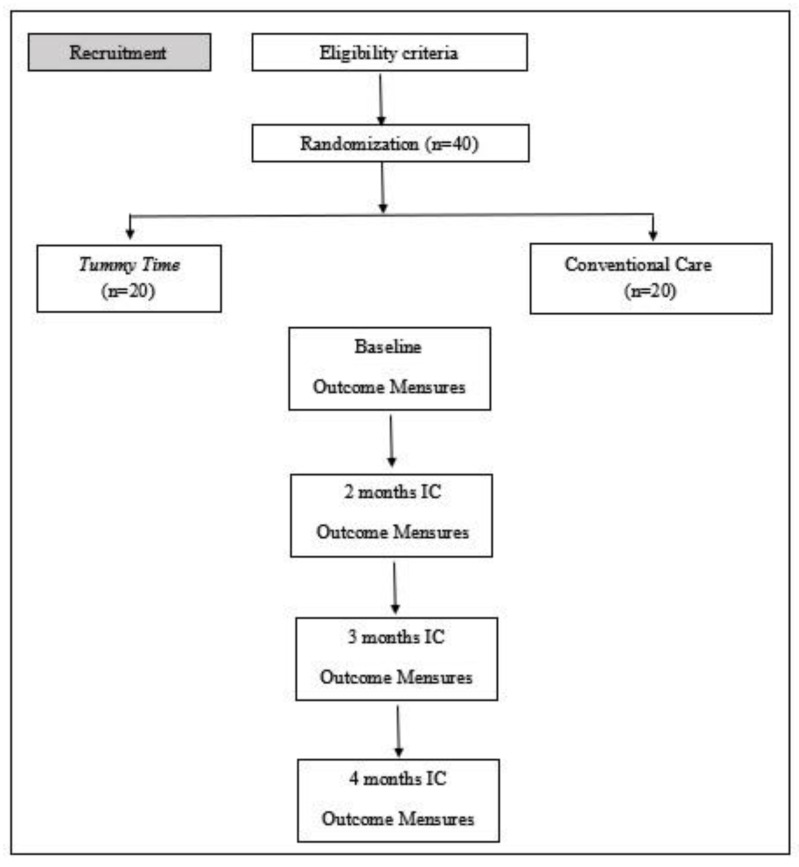
Study schema.

## Data Availability

All data generated or analyzed during this study will be included and are available upon reasonable request from the corresponding author.

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
