# Peer review of "Effects of Prone Positioning on Head Control in Preterm Infants: Randomized and Controlled Clinical Trial Protocol"

_ijerph, 2023, doi:10.3390/ijerph20032375_

Round 1

Reviewer 1 Report

General Overview: this is an interesting protocol of a study with the aim of assessing the effects of prone positioning (tummy time) on cervical extension (angular kinematics and time) and gross motor function in preterm infants. The intervention can be done by parents and is easy to implement with the help of an illustrative booklet.

The introduction shows the importance of the procedure. Study setting, inclusion and exclusion criteria and randomization are well described. Additionally, the flowchart of the study development, ethical aspects and statistical analysis are well stated.

Major concerns: the sample size could be described better. Seems very small, based only in an expected difference in average head elevation angle for repeated measures. It is important to consider the secondary outcome (gross motor function).

Another topic to explain better is concerning the roles of researchers, because it is understood that there is one researcher in charge of allocation concealment, other doing patients’ follow-up, other assessing patients and finally, one researcher for the statistical analyses. How is it coordinated that the investigators in charge of the evaluations and data analyses do not know the allocation to the intervention? Who performs the evaluations is a single investigator? If not a single investigator, how are the evaluations standardized? how is the training and the degree of agreement between evaluators?General Overview: this is an interesting protocol of a study with the aim of assessing the effects of prone positioning (tummy time) on cervical extension (angular kinematics and time) and gross motor function in preterm infants. The intervention can be done by parents and is easy to implement with the help of an illustrative booklet.

The introduction shows the importance of the procedure. Study setting, inclusion and exclusion criteria and randomization are well described. Additionally, the flowchart of the study development, ethical aspects and statistical analysis are well stated.

Major concerns: the sample size could be described better. Seems very small, based only in an expected difference in average head elevation angle for repeated measures. It is important to consider the secondary outcome (gross motor function).

Another topic to explain better is concerning the roles of researchers, because it is understood that there is one researcher in charge of allocation concealment, other doing patients’ follow-up, other assessing patients and finally, one researcher for the statistical analyses. How is it coordinated that the investigators in charge of the evaluations and data analyses do not know the allocation to the intervention? Who performs the evaluations is a single investigator? If not a single investigator, how are the evaluations standardized? how is the training and the degree of agreement between evaluators?

Reviewer 2 Report

Thank you very much for providing me the opportunity to review this interesting and well-written manuscript which is a study protocol of a randomized clinical trial on the effects of prone positioning (30 min per day) on head control in preterm infants. I have only some minor remarks:

-          Will the tummy time (30 min per day) be performed from baseline to the last assessment (i.e. four months of corrected age)?

-          Page 2, line 90: "usual" seems to be "unusual"

-          Page 6, line 203: "home discharge" should be "hospital discharge"

-          While the duration of tummy mode is 30 min in the text, it is 20 min in the supplementary material (i.e. "Start with short periods of 2-3 minutes until you complete 20 minutes daily of Tummy Time")

Reviewer 3 Report

A well-thought-through, design, and clearly presented, thanks.

Line 38: ...to be encouraged in the use of...
